# Loop Closure Detection Based on Multi-Scale Deep Feature Fusion

**Baifan Chen, Dian Yuan ***, **Chunfa Liu and Qian Wu**

School of Automation, Central South University, Changsha 410083, China; chenbaifan@csu.edu.cn (B.C.); 15084843844@163.com (C.L.); wuqian945@csu.edu.cn (Q.W.)
* Correspondence: dianyuan@csu.edu.cn; Tel.: +86-185-7310-0034

**Abstract:** Loop closure detection plays a very important role in the mobile robot navigation field. It is useful in achieving accurate navigation in complex environments and reducing the cumulative error of the robot's pose estimation. The current mainstream methods are based on the visual bag of word model, but traditional image features are sensitive to illumination changes. This paper proposes a loop closure detection algorithm based on multi-scale deep feature fusion, which uses a Convolutional Neural Network (CNN) to extract more advanced and more abstract features. In order to deal with the different sizes of input images and enrich receptive fields of the feature extractor, this paper uses the spatial pyramid pooling (SPP) of multi-scale to fuse the features. In addition, considering the different contributions of each feature to loop closure detection, the paper defines the distinguishability weight of features and uses it in similarity measurement. It reduces the probability of false positives in loop closure detection. The experimental results show that the loop closure detection algorithm based on multi-scale deep feature fusion has higher precision and recall rates and is more robust to illumination changes than the mainstream methods.

**Keywords:** loop closure detection; convolutional neural network; spatial pyramid pooling

## 1. Introduction

Loop closure detection has become a key problem and research hotspot in the field of mobile robot navigation, particularly in simultaneous localization and mapping (SLAM), because it can reduce the cumulative error of robot pose estimation and achieve accurate navigation in large-scale complex environments. Vision-based loop closure detection, also called visual place recognition, is when the robot identifies the places that have been visited before with images provided by the vision sensor during the navigation. For example, assume there are two images captured at the current time and at an earlier time, the problem of loop closure detection is to judge whether the places at the two moments are the same according to the similarity of these two images. Correct loop closure detection can add an edge constraint in the pose map to help optimize robot motion estimation further and build a consistent map. Wrong loop closure detection will lead to the failure of map building. Therefore, a good loop closure detection algorithm is crucial for consistent mapping and even for the entire SLAM system.

At present, the mainstream methods of visual loop closure detection are based on the Bag of Words (BoW), which cluster the visual features into some "words" and then describe an image in the form of a "words" vector. Thus, the visual loop closure detection problem is transformed into a similarity measure problem with the word vectors of the two images. However, the visual features in the BoW are all artificially designed by researchers in the field of computer vision, and they all belong to the low-level features and are sensitive to illumination changes. With the advent of various visual sensors, different visual features are designed based on the different characteristics of the sensors. However, the design of a new visual feature is often very difficult. In recent years, deep learning

methods have developed rapidly. They start from the raw data of the sensor and automatically extract the abstract information of the data through a multi-layer neural network. Compared with traditional image processing, deep learning networks use multiple convolutional layers to extract features and use pooling layers to select features. The extracted image features are more advanced and abstract than traditional artificial visual features. Convolutional Neural Network (CNN) has been widely applied in image retrieval and image classification.

Considering the similarity between visual loop closure detection and image classification (they both need to extract the features of the image, and then complete the related tasks based on the extracted features), this paper applies CNN to loop closure detection and proposes a loop closure detection algorithm based on multi-scale deep feature fusion. The algorithm includes three modules: feature extraction layer, feature fusion layer and decision layer (as shown in Figure 1). We selected the first five convolutional layers of the pre-trained AlexNet network on the ImageNet dataset as the feature extraction layer, which can extract more advanced and more abstract features. In the feature fusion layer, we designed a multi-scale fusion operator with spatial pyramid pooling (SPP) [1] to fuse the deep features with different receptive fields and create a fixed length representation of an image. Finally, in the decision layer, we developed a similarity measurement method by calculating the distinguishability weight of features, which helps reduce the probability of false positives in loop closure detection. The results show that the loop closure detection algorithm has a high precision and recall rate. They also verify the algorithm's robustness to illumination changes.

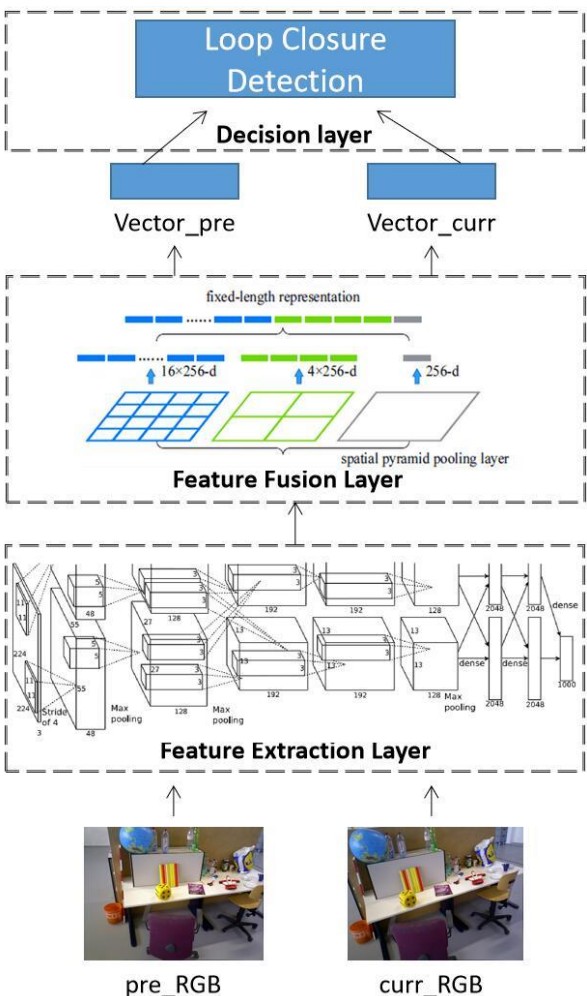

**Figure 1.** The framework of visual loop closure detection algorithm based on multi-scale deep feature fusion.

## 2. Related work

In recent years, scholars have done much research in the direction of loop closure detection algorithms based on vision. The classical algorithms can be roughly divided into two categories: the method based on the BoW (Bag of Word) [2] and the method based on the global descriptor. The first method extracts local features from the scene image and clusters them into multiple "words". Then the whole image is described in the form of vectors based on these "words". Thus, the visual loop closure detection problem is transformed into a similarity measure problem of the description vectors of the two images. BoW is the mainstream method for loop closure detection. A key problem of the BoW method is how to select local features of the image. The common feature points are SIFT [3], SURF [4] and ORB [5]. For example, Mei et al. [6] used the FAST [7] operator to extract the key points and then used the SIFT [3] as feature descriptor. Newman et al. [8] extracted FAST [7] key points and then calculated the descriptors using BRIEFF [9]. For the general case, the images described by the BoW can be compared one-to-one by the histogram or Hamming distance, and the closed loop is detected when the distance is less than a certain threshold. However, in a large-scale scene, search speed is very important, and some researchers have begun to apply the word tree to do efficient loop closure detection. Cummins et al. [10,11] applied Chow-Liu tree approximation to describe the correlation between words and words, and then proposed the classic FAB-MAP method. Glover et al. [12] made public the FAB-MAP development kit based on the work of Cummins et al., which provided convenience for researchers. Maddern et al. [13] proposed the CAT-SLAM method based on FAB-MAP, which combines loop closure detection with a local metric pose filter. Compared with FAB-MAP, the loop closure detection of CAT-SLAM is better. For the second method, the main idea is to describe the entire image with a global descriptor. Ulrich et al. [14] proposed that color histograms provide a compact representation of an image, which results in a system that requires little memory and performs in real-time. But it is very sensitive to changes in illumination. Dalai et al. [15] used histograms of oriented gradients (HOG) as the feature descriptor of the image, which gave very good results for person detection in cluttered backgrounds. GIST [16] had been demonstrated to be a very effective conventional image descriptor, capturing the basic structure of different types of scenes in a very compact way. Based on this, Murillo et al. [17] utilized global gist descriptor computed for portions of panoramic images and a simple similarity measure between two panoramas, which is robust to changes in vehicle orientation, while traversing the same areas in different directions.

Both of the two methods have their own advantages and disadvantages. Furgale et al. [18] proved that the global descriptor method is more sensitive to the camera pose than the BOW method. Milfold [19] and Naseer [20] proposed that the global descriptor method is more robust in the case of illumination changes. Therefore, some researchers have considered combining the two methods and proposed a method of using scene signatures. For example, McManus et al. [21] presented an unsupervised system that produces broad-region detectors for distinctive visual elements, which improved the accuracy of detection. However, the features used in these methods are low-level features and designed artificially in the field of computer vision. They are sensitive to the influence of light, weather and other factors, so these algorithms lack the necessary robustness.

With the disclosure of large-scale datasets (such as ImageNet [22]) and the upgrading of various hardware (such as GPU), deep learning has developed rapidly in recent years. Deep learning can extract abstract and high-level features of the input image through multi-layer neural networks, which is more robust to changes in environmental factors [23,24]. Therefore, it has been widely used in image classification [25] and image retrieval [26]. Considering that visual loop closure detection is similar to image classification and image retrieval, researchers have tried to apply deep learning to loop closure detection. Gao et al. [27,28] took advantage of Autoencoder to extract image features and used the similarity measurement matrix to detect closed loops, which got high accuracy on public datasets. He et al. [29] applied FLCNN (fast and lightweight convolutional neural networks) to extract image features and calculate the similarity matrix, which further improved the real-time and accuracy of loop closure detection. Xia et al. [30] extracted image features by PCANet, and proved that these features

are superior to traditional manual design features. Hou et al. [31] used PlaceCNN to extract image features for loop closure detection, which got high accuracy even when the light changed. However, these methods relied on local deep features and ignored the scale problem.

## 3. Loop Closure Detection Algorithm Based on Multi-Scale Deep Feature Fusion

Different from traditional image classification, visual loop closure detection needs to determine whether the two moments are at the same location according to the similarity between the picture collected at the current time and the one taken earlier. Therefore, the algorithm in this paper has paired input, which corresponds to two branches in the algorithm as shown in Figure 1. The algorithm is divided into three layers: feature extraction, feature fusion and decision. The feature extraction layer extracts the deep feature of the input images. The feature fusion layer does multi-scale fusion and normalization of extracted features. The decision layer uses the fusion feature to detect the loop closure. The two branches of the algorithm are identical in the feature extraction layer and the feature fusion layer structure.

### 3.1. Feature Extraction Layer

The feature extraction layer is composed of two identical CNNs to extract features for two inputs separately. Compared to the early CNN, AlexNet [27] uses a much deeper network model to acquire features. Additionally, it adds modules such as the ReLU activation function, local response normalization (LRN), Dropout, etc., which can reduce the risk of over fitting. Moreover, it takes advantage of multi-GPU to improve the training speed of the network model. In view of these advantages, this paper refers to AlexNet.

The network model has a total of eight layers, consisting of five convolution layers and three fully connected layers. Only the first five layers of AlexNet are needed, as shown in Figure 2. Here, we set the input of the feature extraction layer to be an RGB image of $227 \times 227 \times 3$, and obtained 256 feature maps with a size of $6 \times 6$ through five convolution layers. (The output data size of each layer is indicated in Figure 2). The convolution layer contains the ReLU activation function and LRN processing, as well as max-pooling. Among them, LRN, which draws on the idea of "lateral inhibition" in neurobiology, is used to locally suppress neurons. When the activation function is ReLU, this "lateral inhibition" is very useful. It can prevent the model from over-fitting prematurely and speed up the training of the model. Its calculation formula is

$$b_{x,y}^i = a_{x,y}^i / \left( k + \alpha \sum_{j=\max(0,i-n/2)}^{\min(N-1,i+n/2)} \left( a_{x,y}^i \right)^2 \right)^\beta, \tag{1}$$

where $a_{x,y}^i$ denotes the value of the *i*-th convolution kernel after applying the ReLU activation function at position (x, y); n is the number of convolution kernels adjacent in the same position, and N represents the total number of convolution kernels. $k, n, \alpha$ and $\beta$ are tunable parameters; we set $k = 2, n = 5, \alpha = 0.0001, \beta = 0.75$ according to the empirical value.

### 3.2. Feature Fusion Layer

An RGB image can obtain several feature maps describing different deep features from features extraction layer. For general image classification tasks, a fully connected layer is usually added to weight and sum all the feature maps to obtain the classification result. However, each feature map only corresponds to a small area in the original input image, and its receptive field is small. The deep feature is a local feature. For the loop closure detection problem, we prefer to have different scales of features because this helps to accurately determine whether two images belong to the same scene. In addition, AlexNet [27] requires the fixed-size of the input image to be $227 \times 227 \times 3$. When the size of the input does not meet the requirements, cutting or compressing must happen, which causes the loss or distortion of some image information.

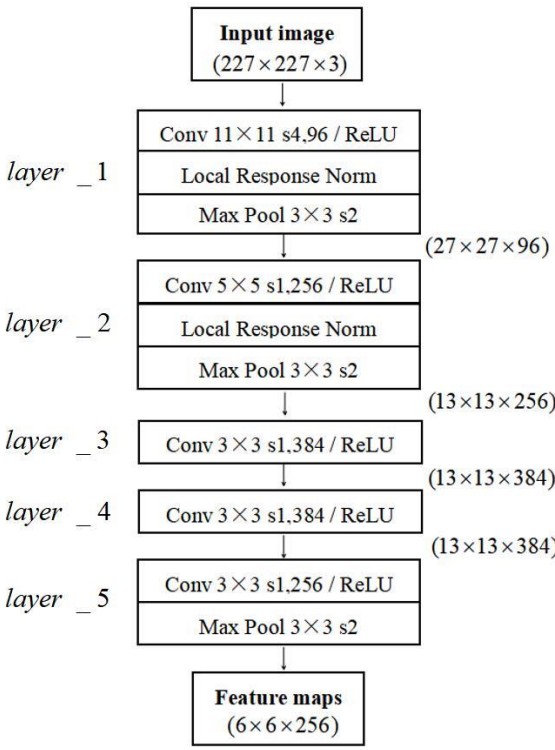

**Figure 2.** Structure of feature extraction layer.

To overcome this problem, we use spatial pyramid pooling (SPP) [1] to fuse multi-scale deep features. SPP divides the feature map into small patches of different sizes by different scales, and then gets each patch's feature. The receptive fields to the input images of these small patches are different, which means the patches correspond to different areas of the input image. Finally, the features extracted from the small patch of different sizes are combined to achieve the fusion of multi-scale features. In addition, the SPP layer can get a fixed-size output and eliminate the restriction that the input image size must be fixed.

Taking a single branch as an example, the feature fusion layer is shown in Figure 3. Here, the input of feature fusion layer is the output from feature extraction layer. After the SPP layer, we can obtain the fixed-length representation of the features, which is finally sent to softmax.

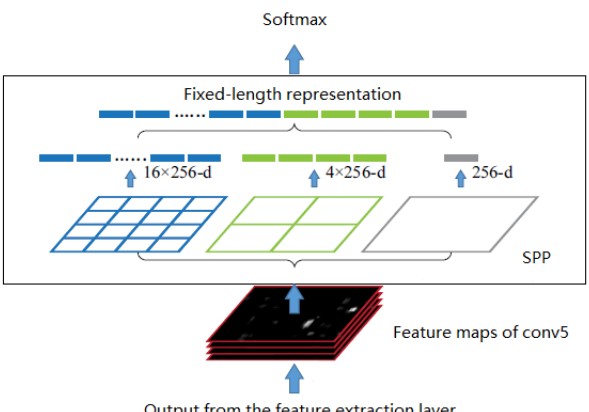

**Figure 3.** Feature fusion layer.

As seen in Figure 3, SPP divides each feature map by using three different scales $4 \times 4$, $2 \times 2$, $1 \times 1$. Each scale corresponds to one layer in the SPP; three scales indicate that the SPP has three layers colored as blue, green and gray. For example, the $4 \times 4$ scale divides a feature map into $4 \times 4$

small patches, and then extracts a feature from each small patch, so each feature map can extract 16 features. There are 256 feature maps as output from the feature extraction layer, and altogether there are $16 \times 256$ features (blue parts in Figure 3). The features extracted by the different scales of the SPP are concatenated together to obtain a fixed-length feature vector. In order to normalize the results of the SPP, we add a SoftMax layer. Assuming that the feature vector output by the SPP layer is $Z \in R^{n \times 1}$, the output of the SoftMax layer is:

$$Y = \left[ \frac{e^{Z_1}}{\sum_{j=1}^{n} e^{Z_j}}, \cdots, \frac{e^{Z_n}}{\sum_{j=1}^{n} e^{Z_j}} \right]^T, Y \in R^{n \times 1}, \quad (2)$$

The number of SPP layers affects the output size of the SoftMax layer. If the number of layers is different, the performance of the corresponding algorithm will be different. In the experiment part, the effects of SPP in different layers on the performance of the algorithm are discussed.

*3.3. Decision Layer*

The decision layer is in charge of loop closure detection. Suppose the inputs of the two branches are Image_1 and Image_2, respectively, and their feature vectors of the SoftMax layer are $f_1$ and $f_2$. The vector dimension is determined by the number of SPP layers and the scale of each layer. Assuming that the dimensions of $f_1$ and $f_2$ are $N$, the similarity of two inputs can be calculated by Equation (3).

$$S_1(f_1, f_2) = 1 - \sum_{i=1}^{N} (f_{1i} - f_{2i}), \quad (3)$$

where $f_1$ and $f_2$ represent the i-th dimension of the feature vector. Setting a threshold T, when $S_1(f_1, f_2) > T$, it means that Image_1 and Image_2 correspond to the same place and a closed loop is detected. Equation (3) treats each feature node fairly, but the distinguishability of each feature node is different. Features such as walls and ground are more common in scenes and their distinguishability is relatively small, while the traffic signs are more distinguishable. If the feature nodes with different distinguishability are treated fairly, more false positives (different places in similar scenes) will occur when detecting the loop closure. Considering this character, we add a weight to each feature node and modify Equation (3) as Equation (4). The weight's value indicates the distinguishability of the feature to the scene.

$$S_1(f_1, f_2) = 1 - \sum_{i=1}^{N} \delta_i (f_{1i} - f_{2i}), \quad (4)$$

where $\delta_i$ represents the weight of the *i*-th feature node, and the larger the value of $\delta_i$, the greater the distinguishability of its corresponding feature in the scene. The weights can be learned by training and accord with Gaussian distribution. The value of $\delta_i$ is calculated as follows:

$$\delta_i = \exp(-\frac{(\overline{h}_i - u)^2}{2\delta^2}), \quad (5)$$

where $\overline{h}_i$ represents the average response of the *i*-th feature node. If the average response of a feature node is larger, such features are more common (such as ground and sky), and the $\delta_i$ is smaller. If the average response of a feature node is smaller and lower than the mean value of $u$, such features are not common (such as noise), and $\delta_i$ is also smaller. When $\overline{h}_i$ is near the mean value of $u$, the distinguishability is relatively large, and the corresponding weight value is also relatively large. After the network model is trained, the value of $\overline{h}_i$ can be calculated and retained by the test. $u$ and $\delta$ are tunable parameters and set according to experience.

## 4. Parameter Training

### 4.1. Training Method

Since SPP in the feature fusion layer is a special maxing pooling layer, it has no parameters to be trained. Here we only need to train the parameters of the convolution network in the feature extraction layer. The algorithm in this paper consists of two branches with the same structure in the feature extraction layer and the feature fusion layer. For the loop closure detection problem, its positive sample indicates the same location, and the negative sample indicates non-loop closure. Loop closure always happens at different locations and with very few times, which means a large number of classes and small labeled samples. Taking these into consideration, this paper uses the Siamese [32] model to train.

The Siamese network is mainly used in the field of face recognition can solve the classification problem with small sample data well. Figure 4 shows the training model.

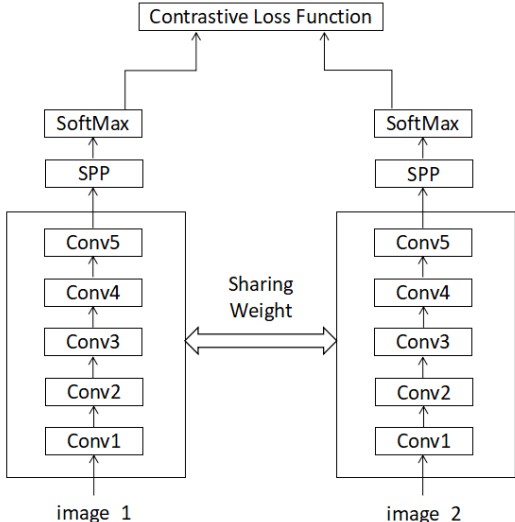

**Figure 4.** Parameter training model for feature extraction layer.

Contrastive loss function is used here. For the $i$-th pair of samples, we assume that the feature vectors of the SoftMax layer are $f_{i1}$, $f_{i2}$ respectively, and the contrastive loss function is:

$$L = \frac{1}{2M}\sum_{i=1}^{M}[y_i d_i^2 + (1 - y_i)\max(m\arg in - d_i, 0)^2], \tag{6}$$

where $M$ is the number of sample pairs; $d_i = \| f_{i1} - f_{i2} \|_2$ is the Euclidean distance of the two samples in the feature space, and $y_i$ is the label of the $i$-th pair of samples. $y = 1$ is the positive sample, which means that the two images are similar and form a loop closure; $y = 0$ represents a negative sample, which means that the similarity of the two pictures is small and does not form a loop closure. The threshold $margin$ is set to 1 in the experiment. When $y = 1$ and the loss function is $L = \frac{1}{2M}\sum_{i=1}^{M}y_i d_i^2$, if their Euclidean distance d in the feature space is large, the current training model is not good and its loss value increases. When $y = 0$ and the loss function is $L = \frac{1}{2M}\sum_{i=1}^{M}[\max(m\arg in - d_i, 0)^2]$, if their Euclidean distance in the feature space is small, the loss value of the model will become large. The contrastive loss function can express the matching degree of paired samples, and it can be used for the model training.

### 4.2. Model Training

We use the AlexNet model pre-trained by the ImageNet dataset on Caffe [33], and set the parameters of the first five convolutional layers as the initial values of the parameters in the feature extraction layer. Meanwhile, we use the Matterport3D dataset to fine tune the model. The Matterport3D

dataset is the world's largest public 3D dataset from 3D scanning solution provider Matterport. It is a large-scale RGB-D dataset containing 10,800 panoramic views from 194,400 RGB-D images of 90 building-scale scenes. The dataset was acquired by a Pro 3D camera. The camera rotates around the center of gravity at each sampling point, and samples the images at six rotation positions, each of which corresponds to 18 sets of pictures. This special way of data acquisition makes the camera cover a wider range of angles, and it provides many loop closure data. Figure 5a–f are some examples of RGB images in the MatterPort3D dataset.

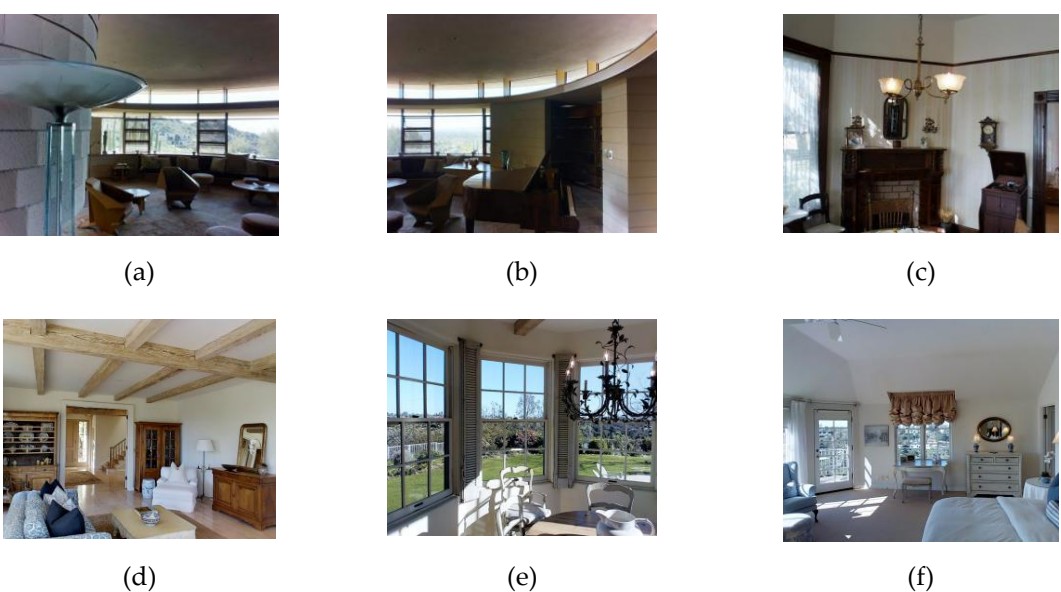

(a)　　　　　　　　　　　　　　(b)　　　　　　　　　　　　　　(c)

(d)　　　　　　　　　　　　　　(e)　　　　　　　　　　　　　　(f)

**Figure 5.** Examples of an RGB graph in the Matterport3D dataset.

We selected 30,000 pairs of positive samples (loop closure) and 30,000 pairs of negative samples (non-loop closure) in the Matterport3D data set. Then by some data enhancement means such as rotating and translating, the positive and negative samples were each doubled to 60,000. Of these, 10,000 pairs of positive samples and 10,000 pairs of negative samples were selected as test sets, and the other 50,000 pairs of positive samples and 50,000 pairs of negative samples were used for training. Some researchers have found that multi-scale training can improve the accuracy of the network containing SPP layers in the tasks of image classification and object detection. In view of this, this paper uses three different scales of data to train the model: $1280 \times 1024$, $227 \times 227$ and $180 \times 180$. The latter two data are cropped from the original image. In this way, the number of our training set is 300,000, including 150,000 pairs of positive samples and 150,000 pairs of negative samples.

The deep learning framework Caffe requires a fixed size of input images; therefore, we used a combination of single-scale and multi-scale to train the model. In one epoch of model training, single-scale images are input, while in each different epoch of model training, the input batches are of different scales.

In addition, in order to analyze the influence of different SPP layers in the feature fusion layer on the performance of the algorithm, three different SPP are used: a one-layer SPP with a scale of $1 \times 1$, a two-layer SPP with a scale of $1 \times 1$, $2 \times 2$, and a three-layer SPP with a scale of $1 \times 1$, $2 \times 2$, $4 \times 4$. For these three cases, we build three models and train them separately.

## 5. Experiment

In order to verify the performance and effectiveness of the algorithm, we tested the effects of different layers of SPP, the influence of the similarity measurement method in visual loop closure detection and the robustness to illumination changes. An Intel Core i7 processor of 2.8 GHz frequency and an NVIDIA GeForce GTX 1060 with MAX-Q Graphics card were used.

*5.1. Dataset and Labeling*

The dataset was provided by the computer vision group of the Technical University of Munich (TUM) [34]. We used the dataset's RGB-D image sequence, which includes Fr2/rpy, Fr2/large_with_loop, Fr2/pioneer_slam, Fr2/pioneer_slam2 and Fr3/long_office_household. However, the loop closure is not labeled in the TUM dataset and must be manually marked. Because of the 30 Hz/s sample frequency, there are too many images in the same place. Therefore, we selected the key frames with the ground truth of trajectory.

Keyframe selection includes the following steps.

For each image sequence, a key frame list $F = \{\}$ is set, and the first frame is added to $F$.

Sequentially compare the image $f_j$ in the image sequence with the last frame

$$f_i$$

in the key frame list. Assuming that their corresponding poses are $T_j, T_i$ which can be found in the ground truth, the relative translation of the two frames is $M_{j,i} = trans(T_j^{-1}T_i)$, where $trans(\bullet)$ represents the 2 norm of the translational part of the transformation matrix. If $0.1 < M_{j,i} < 1$, the frame $f_j$ is added to the end of the key frame list.

Finally, the loop closure is manually labeled with the selected key frame sequence.

Figure 6a shows the trajectory ground truth of the Fr3/long_office_household sequence, and in Figure 6b red dots are the selected key frames and the black line segments are the loop closure. Figure 6c,d are one pair of loop closure images.

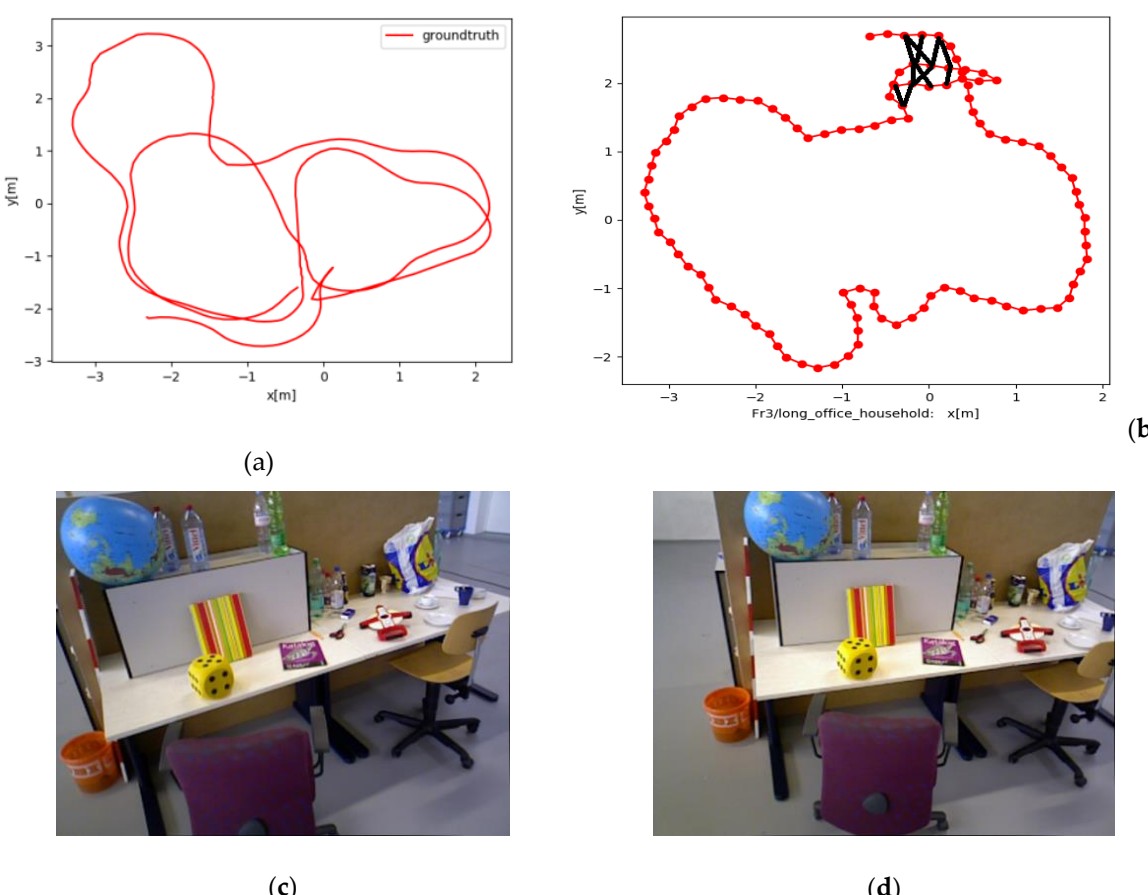

**Figure 6.** Keyframe selection and loop closure example in FR3/long_office_household.

*5.2. Different layers of SPP*

SPP is used in the feature fusion layer to fuse the extracted depth features of different scales. In order to verify how different layers of SPP influence the algorithm results, an SPP with 1 × 1, a two-layer SPP with 1 × 1, 2 × 2, and a three-layer SPP with 1 × 1, 2 × 2, 4 × 4 were used for experiments. The loop closure detection algorithm with these three different SPP layers, noted as spp1, spp12 and spp124, respectively, was compared with FabMap [10].

We selected 689 key frames from the Fr2/rpy, Fr2/large_with_loop and Fr3/long_office_household and labeled 45 loop closure places, which were noted as data_1. The experimental results of the data_1 are shown in Figure 7. It can be seen from Figure 7 that the P-R curves of spp1, spp12 and spp124 are basically on the upper right of the coordinate system, which means they have higher precision and recall rates. Our method can reach 100% precision at 50% recall. In addition, spp124 is better than spp12 and spp1, and spp12 is better than spp1. When the recall rate is 100%, spp124 reaches as high as 78% precision; spp12 is about 62% precision, and spp1 is less than 50% precision. It shows that the greater the layers of SPP, the higher the precision and recall rate of the algorithm.

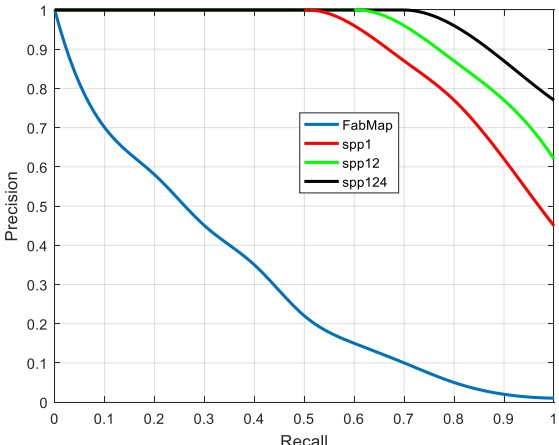

**Figure 7.** P-R curves on data_1.

The above algorithm was also tested in the Fr2/pioneer_slam and Fr2/pioneer_slam2 image sequences which are sampled in a very empty indoor environment and have many similar scenes, such as walls, boards and ground. We selected 435 key frames from the two sequences and labelled 20 loop closure places, which were noted as data_2. The test results on data_2 are shown in Figure 8.

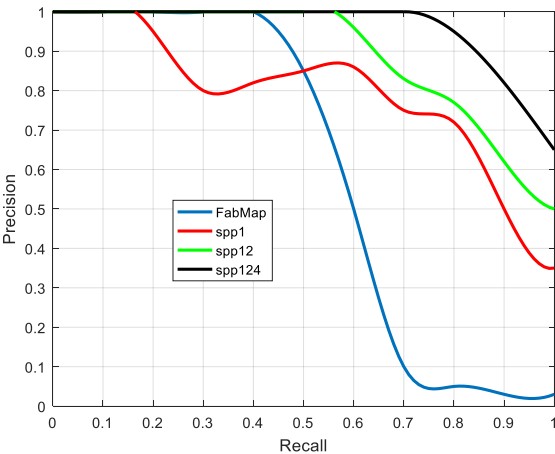

**Figure 8.** P-R curves on data_2.

As can be seen from Figure 8, spp124 has the highest precision-recall rate, followed by spp12. At the 50% recall rate, both spp124 and spp12 can reach 100% precision. For spp1, when the recall rate is less than 50%, the accuracy is higher than the accuracy of FabMap, but when the recall rate is greater than 50%, the accuracy of spp1 is lower than that of FabMap. It is worth mentioning that the performance of all algorithms on data_2 is worse than on data_1. Because there are many similar scenarios in data_2, the algorithms got some false positive results. Overall, the depth features extracted by CNN are more suitable for loop closure detection compared with the traditional artificial design visual features, and increasing the number of SPP layers in the feature fusion layer can improve the accuracy and recall rate.

### 5.3. Similarity Measurement

In order to verify the effect of weight adjustment in the similarity measurement, we compared the spp1, spp12, and spp124 methods with these added weight adjustments which were denoted as spp1+, spp12+, spp124+. The former directly uses Equation (3) to calculate similarity, and the latter uses Equation (4). In order to calculate $\delta_i$ from Equation (5), we selected 100,000 RGB images from the Matterport3D data set and calculated the average response of each node of the SoftMax layer as $h_i$. Since two branches are identical and share their parameters, only one branch needed to be calculated. The tunable parameter was set as $u = 0.5, \delta = 0.1$. The results of data_1 and data_2 are shown in Figures 9 and 10, respectively.

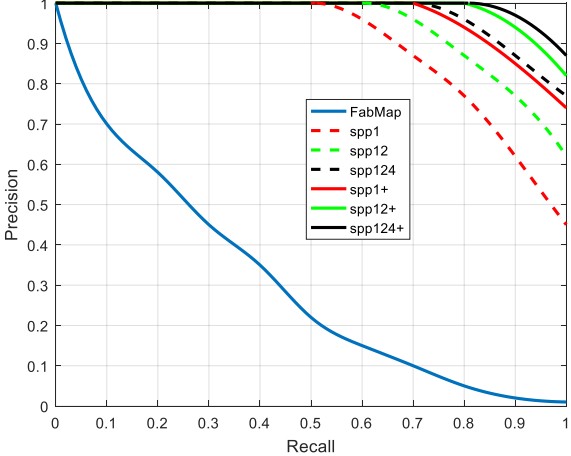

**Figure 9.** P-R curves of seven algorithms on data_1.

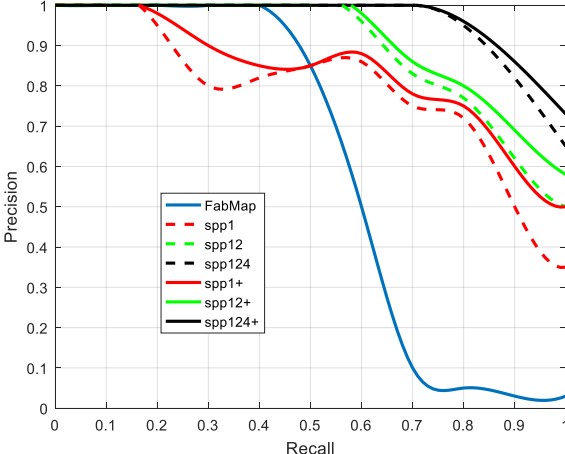

**Figure 10.** P-R curves of seven algorithms on data_2.

At a high recall rate, spp1+, spp12+ and spp124+ have higher precision than spp1, spp12 and spp124. When considering feature distinguishability, the visual loop closure detection algorithm reduced the probability of false positives and improved precision.

### 5.4. Illumination Changes

Illumination changes are the most critical factors affecting the visual loop closure detection. In order to test the robustness of the algorithm to illumination changes, we collected image sequences with different illuminations. We fixed a trajectory and then sampled image sequences at 12:30, 15:00, 17:30 and 19:00. Finally, four image sequences were obtained and named as VS1230, VS1500, VS1730 and VS1900 respectively, as shown in Figure 11. Each image sequence contained 200 frames and 20 closed loops.

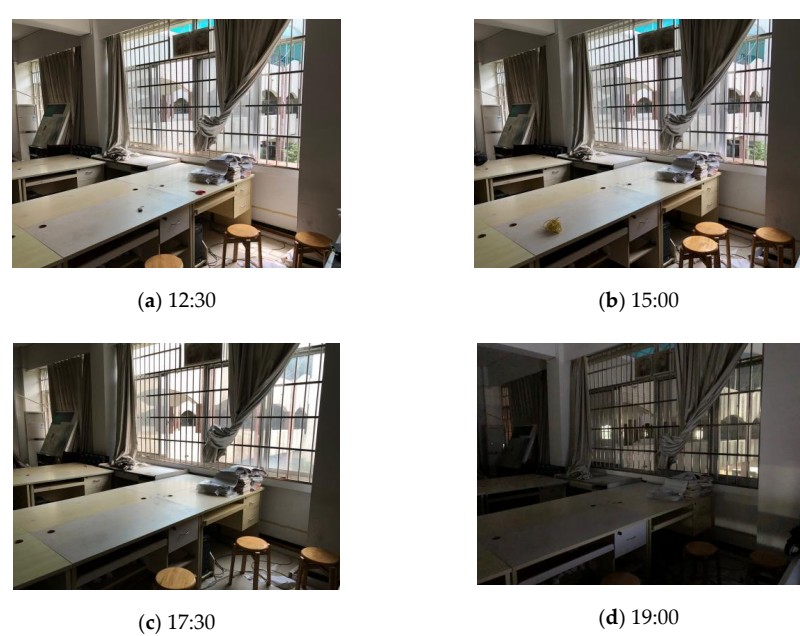

(**a**) 12:30  (**b**) 15:00

(**c**) 17:30  (**d**) 19:00

**Figure 11.** Sample images taken at four different times.

The tests were performed on these four image sequences, and the results are shown in Figures 12–15. Since the algorithms of spp1+, spp12+, spp124+ have been proven to be outstanding in 5.3, here we only compared them with FabMap.

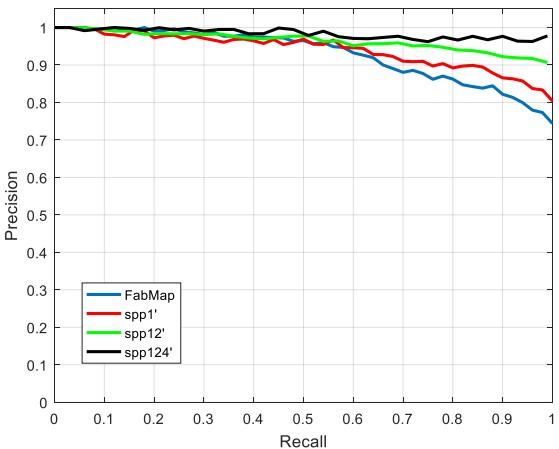

**Figure 12.** P-R curves of four algorithms on VS1230.

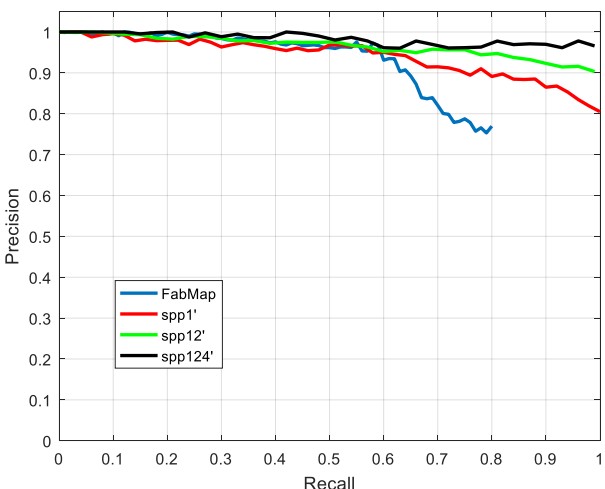

**Figure 13.** P-R curves of four algorithms on VS1500.

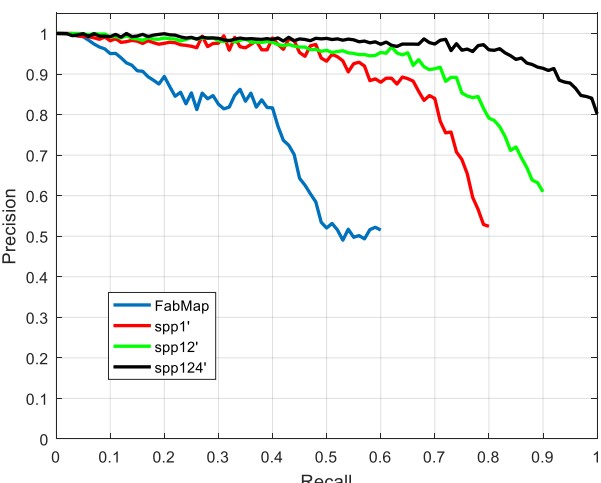

**Figure 14.** P-R curves of four algorithms on VS1730.

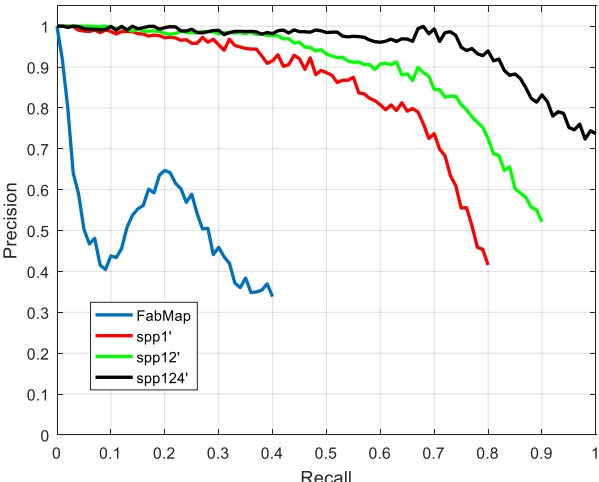

**Figure 15.** P-R curves of four algorithms on VS1900.

At 12:30, the P-R curves of the four algorithms are on the upper right of the coordinate system. However, as the time went by, all P-R values of the algorithms decreased, and the FabMap decreased more obviously because the illumination turned to dim. Especially in the VS1730 and VS1900 image

sequences, the accuracy of FabMap was drastically lower at high recall rates. In the VS1900 image sequences, at 100% recall rate, the precision of spp124+ was still as high as 70% or more. This shows that the proposed algorithm is more robust to illumination changes than the traditional BoW method.

In general, for RGB-D SLAM, researchers prefer higher precision because the wrong closed loop leads to a completely wrong pose graph. Therefore, we also did statistical analysis about the average precision of spp1+, spp12+, spp124+, and FabMap on the four image sequences VS1230, VS1500, VS1730, and VS1900.

As can be seen from Figure 16, the average precision of the four algorithms was above 90% at 12:30, and at 17:30, only spp124+ and spp12+ were still above 90% precision. At 17:30, the average precision of FabMap is only 70%. At 19:00, the differences are more obvious. At this time, the average precision of spp124+ was still as high as 90%, and the other three were below 90%, especially FabMap which had an average accuracy of only 42%. It can be concluded that the algorithm of this paper is more robust than the traditional BoW method when the illumination changes.

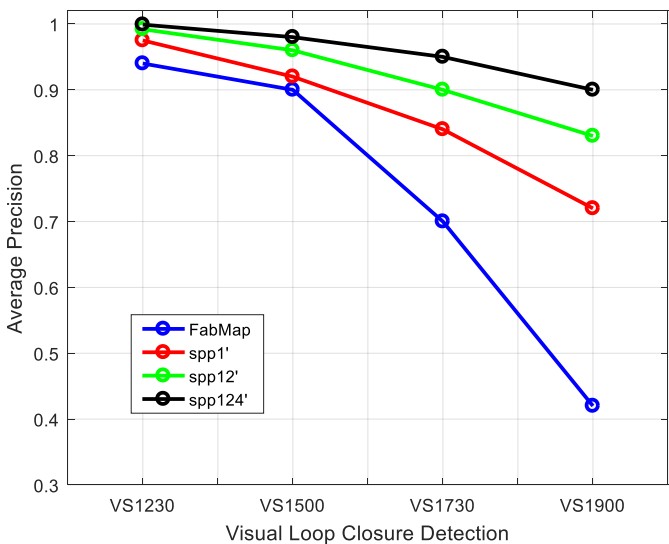

**Figure 16.** Average precision of spp1+, spp12+, spp124+, and FabMap on four image sequences.

## 6. Conclusions

This paper has proposed a loop closure detection algorithm based on multi-scale deep feature fusion. Our proposed algorithm has a higher accuracy and recall rate than the traditional BoW method, and it has better robustness to illumination changes. The key ideas that allowed us to achieve this efficiency are as follows. First, we used CNN to extract features. The features extracted in this way are more advanced and abstract, and have better robustness to illumination changes. Second, we used SPP to fuse the extracted features. By setting the multi-layer SPP with different receptive field sizes to fuse the different scale features in the image, it is more conducive to detecting loop closure. Last but not least, in the similarity calculation process of the decision layer, we added a weight to each feature node according to the distinguishability of the feature nodes to the scene. We have verified the benefits of the above points with experiments. In fact, the neural network used only plays the role of extracting more abstract features. Later, we will consider introducing the difference between the two image features in the middle layer of the deep network, and using the difference feature to detect loop closure.

**Author Contributions:** Conceptualization, B.C. and C.L.; Data curation, D.Y.; Formal analysis, B.C. and D.Y.; Investigation, Q.W.; Methodology, B.C. and C.L.; Project administration, B.C.; Resources, Q.W.; Validation, D.Y.; Writing—original draft, D.Y.; Writing—review & editing, B.C. and D.Y.

**Funding:** This research was supported by the National Natural Science Foundation of China under Grant No. 61403423, the Natural Science Foundation of Hunan Province of China under Grant No. 2018JJ3689, National

Key Research and Development Plan (2018YFB1201602) and Science and Technology Major Project of Hunan Province of China under Grant No. 2017GK1010.

**Conflicts of Interest:** The authors declare no conflict of interest.

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
