# Peer review of "Loop Closure Detection Based on Multi-Scale Deep Feature Fusion"

_applsci, doi:10.3390/app9061120_

Round 1

Reviewer 1 Report

The authors propose a method of loop closure detection based on images acquired by the mobile robot. The style of the paper has no problems. In spite of the proposed method introduced in the paper could be interesting, it should be necessary the authors take into account the following considerations:

There are in the scientific literature other proposed methods to describe the entire image through global descriptors, for example HOG, GIST, ... These global descriptors are analysed under different situations (ilumination changes, dynamic environments, ...) and as in panoramic images as in omnidirectional images. Such methods and analysis should be taking into account in section 1 and 2 (Introduction and Related work)

The output of feature extraction layer should be specified. In line 164 appears that 'there are 256 feature maps as output from the feature extraction layer', but this should be detailed in section 3.1

In line 158 appears 'the feature fusion layer is shown in Figure 3. Here we set the input of the feature extraction layer to be an RGB image of 227x227x3'. In such figure does not appear the input of the feature extraction layer.

Titles of section 4.1 and 4.2 are indistinguishable 'Training model' and 'Model training'. I suggest change one of them

It is not clear how different parameters (for example u and sigma in line 204) are tunable.

Although the experiments are adequate, it would be convenient to perform an experiment where dynamic changes or some occlusions occur in the scenes

Author Response

We upload it as a Word file. Thanks.

Reviewer 2 Report

First of all nice research. The topic is well described  and built up. 

Some comments: 

- 4.2 Model Training (line 237) is not clear to me. What is considered to be a positive / negative sample? I can distinguish two scenarios, one where the positive samples are same-location-neighbouring-angle images, but this gives rise to the question that the images can actually be very different. The second scenario is where the positive samples are similar-angle-different-location. The latter of which would make more sense to me.   

- 4.2 Also curious how many images are in the Matterport3d dataset and why 30.000 were selected. 

- Does the performance decrease when increasing the size of the SPP above 4x4? 

- Update the images to better fit the page, especially fig 1. is unreadable unless zoomed in. 

Author Response

We upload it as a Word file.Thanks.

Round 2

Reviewer 1 Report

The reviewers have followed the suggestions previously indicated.

In my opinion the paper should be accepted for publication.